# Methods for Pacific Outer Continental Shelf Wind Characterization for Offshore Wind Development

Macy Frost Chang<sup>1</sup>, Raghavendra Krishnamurthy<sup>2</sup>, and Fotini Katopodes Chow<sup>1</sup> <sup>1</sup>Civil and Environmental Engineering, University of California, Berkeley, California, USA <sup>2</sup>Pacific Northwest National Laboratory, Washington, USA **Correspondence:** Fotini K. Chow (tinakc@berkeley.edu)

**Abstract.** The rapid development of the U.S. offshore wind industry has necessitated accurate assessment and prediction of offshore wind profiles to manage and forecast generated power. In 2021, the Bureau of Ocean Energy Management (BOEM) identified two areas on the Pacific Outer Continental Shelf (OCS), the Humboldt and Morro Bay Wind Energy Areas (WEAs), as potential locations for offshore wind farms. Although these areas have historically lacked high-quality observations of wind

- characteristics at typical wind turbine hub heights, scientific buoys sponsored by the Department of Energy and deployed by the Pacific Northwest National Laboratory (PNNL) recently made precise hub-height wind data available for the first time in these locations, introducing novel opportunities for model validation and intercomparison. Performances of 100-meter AMSL wind speed prediction are compared between a conventional physical law-based approach, known as the stabilitycorrected logarithmic law (S-C log law), and three machine learning (ML) approaches, known as random forest (RF), Guassian
- process regression (GPR), and long short-term memory neural network (LSTM). Predictor variables for the ML approaches are constrained to only sea surface-elevation measurements, and the ML algorithms are respectively trained and tested between the two separate locations (over a notable extrapolation distance of 631 kilometers) in order to simulate realistic industry applications and preclude model overfitting from biasing performance metrics. The S-C log law and LSTM produce the most accurate predictions, with average root mean squared error (RMSE) of 1.33 m/s and 1.38 m/s respectively. The error metrics
- of all three ML methods generally improve when a longer training time is implemented and when the algorithms are trained and tested at the same location, with many performance metrics of LSTM surpassing those of the S-C log law. The LSTM and GPR techniques overall exhibit similar or improved offshore wind speed prediction capabilities in comparison to the S-C log law, which is a widely accepted wind speed extrapolation method. These ML techniques are more adaptable for wind energy purposes than conventional physical extrapolation laws, as they can be used to predict other wind parameters and
- generate short-term forecasts. The increasing future availability and fidelity of vertical wind profile data from the Pacific OCS will be vital for determining the degrees of ML performance degradation over smaller train-test distances and for evaluating performance and power output metrics of larger offshore turbines.

## 1 Introduction

- Wind energy is a renewable energy technology that is continuing to rapidly expand and evolve across the globe. Wind energy generation has grown by approximately 15% annually over the past decade, making it the current largest source of renewable energy in the United States ESMAP. With record installations of wind turbines worldwide in 2023, global wind power capacity has surpassed 1 terawatt and currently supplies approximately 10% of global electricity demand WWEA (2024). Recent technological advances and stakeholder interest are allowing new areas to become feasible for wind farm construction. Bottommounted and floating wind turbine structures allow for construction and operation of wind farms in offshore locations, where
- wind speeds tend to be higher and more consistent than over land Wang and Wang (2015). The U.S. Bureau of Ocean Energy Management (BOEM) has approved several wind energy lease areas along the Atlantic Coast. Three offshore wind sites on the Atlantic coast are already in operation and demonstrate the economic and energetic feasibility of offshore wind energy siting (BOEM, 2025).

Sections of the Pacific Outer Continental Shelf (OCS), the area of ocean extending to about 200 nautical miles from the U.S.
West Coast, have recently come into consideration for offshore wind development. In May of 2021, plans were announced to develop floating offshore wind farms in two Pacific OCS locations off the coast of Morro Bay and Humboldt County, which together have the potential to provide up to 4.6 GW of electricity, equivalent to powering 1.6 million homes Release (2021). These two locations were titled as "Wind Energy Areas" (WEAs) by BOEM after collaborating with federal agencies, the state of California, local tribes and communities, and the public to consider potential impacts on the environment as well as ocean

resources and commerce. In 2022, the WEAs were split into five subareas for lease and auctioned off to private developers McCoy et al. (2024).

**Figure 1.** The geographic areas of the Humboldt and Morro Bay Wind Energy Areas (WEAs) identified by the U.S. Bureau of Ocean Energy Management. (Images adapted from BOEM.gov, BOEM (a, b))

Along the U.S. West Coast, the impact of atmospheric and oceanographic conditions on offshore wind turbines is largely unknown. Due to the sharp gradient in the bathymetry of the seafloor extending from the coastline, future wind farms will primarily be composed of floating offshore wind turbines. Furthermore, existing high-resolution models for estimating the
wind resource are challenged by complex wind-wave-terrain interactions, large amounts of cloudiness, and shallow atmospheric boundary layers typically observed in this region. Therefore, the development of offshore wind farms requires extensive research of atmospheric and oceanographic factors that affect wind farm layout, predicted power output, and fatigue load estimations. Until recently, only a few sea surface-monitoring buoys scattered along the Pacific OCS have provided direct observational data of offshore meteorological and oceanic characteristics. These buoys measure only surface variables and do not
provide vertical profiles of winds and turbulence throughout the surface atmospheric boundary layer (ABL), which is critical

information needed by wind energy developers.

As of September 2020, a research campaign funded by BOEM and the U.S. Department of Energy (DOE) has made direct, consistent observations of above-surface wind characteristics in the Pacific OCS available for the first time. The Pacific campaign consisted of two specialized research buoys stationed in the Humboldt and Morro Bay WEAs. These buoys measure

- wind speed and direction up to 250 meters above mean sea level (AMSL) as well as a large set of surface variables (Krishnamurthy et al., 2023). This data can be used to assess the wind resource at potential turbine heights in these Pacific OCS sites. It also introduces new possibilities for validating wind prediction models in the Pacific OCS region. More specifically, the observational data is vital for identifying model biases and optimizing the performance of statistical, physical, and intelligent learning wind prediction methods Shao et al. (2021); Sheridan et al. (2024); Liu et al. (2024); Bodini et al. (2024).
- New strategies for wind speed extrapolation from sea surface-elevation data could prove to be more adaptable and sitespecific than relatively complex and computationally-expensive numerical weather prediction (NWP) models (de Montera et al., 2022). Additionally, improved wind modeling accuracy and new intelligent modeling approaches may help to enhance confidence in turbine and wind plant power production estimates, which would have significant impacts on downstream analyses, including grid integration and expansion Ortega et al. (2020); Mahoney et al. (2012); Hasager et al. (2015) as well as
- life-cycle cost analyses of floating offshore wind energy production Jong et al. (2017). At both California WEAs, reanalysis data consistently falls short in capturing the full potential of the rotor-level wind resource (Sheridan et al., 2022). The most significant errors arise during stable atmospheric conditions, when wind speeds surpass 10 m/s, which is critical for peak turbine power generation, and when diurnal the wind speed variation in summer months is poorly represented, leading to substantial errors in estimation of wind energy potential (Sheridan et al., 2024). Considering the recency of the collected Pacific OCS wind
- data, an imminent research need is to test out different wind prediction models and assess which techniques provide the best wind estimates for energy developers (Gaudet et al., 2024; Liu et al., 2024, 2025).

A large suite of wind prediction techniques are in use by the wind industry and could potentially be tested in these offshore areas. One category of techniques is physical modeling, wherein meteorological variables are input into physical law equations to calculate other weather characteristics (Fairall et al., 1996; Edson et al., 2013). Another contemporary group of

75 techniques are intelligent learning methods, which most often use machine learning (ML) architecture to draw complex statis-

tical relationships between input data and subsequently make predictions (Crippa et al., 2021; Optis et al., 2021; Vassallo et al., 2020a).

In this study, prediction methods from both of these categories are tested by comparing their accuracy and adaptability across the Humboldt and Morro Bay WEAs. The hub-height wind characteristics are analyzed to find relationships between these hub-height winds and surface variables, where "hub height" is generalized as 100 AMSL for this study, but could be 80 extended to any height to match newer offshore turbine sizes. These relationships are drawn from analyzing correlations of 100-meter AMSL winds to surface wind speed, air and sea temperature, atmospheric pressure, and temporal characteristics. The primary goal is to compare vertical wind extrapolation performance of a widely-used physical law approach, the stabilitycorrected logarithmic law (denoted as S-C log law), versus the machine learning methods of random forest (RF), Gaussian process regression (GPR), and long short-term memory neural network (LSTM). 85

The three aforementioned ML algorithms were selected for this study due to their varied advantages and shortcomings for wind profiling purposes. RF effectively handles missing values, nonlinear parameters, and outlier data, though its stable nature may limit its effectiveness in capturing stochastic variability in time series data. GPR also handles missing and nonlinear data well and provides confidence intervals for predictions, but the computational intensity required by its nonparametric design

- may limit efficiency for large datasets. LSTM has demonstrated high accuracy for wind speed prediction due to its specific design for retaining long-term patterns in time series data, but it requires more hyperparameter tuning to each specific use case as well as complete datasets. To the authors' knowledge, these three ML algorithms have not been compared within one wind speed prediction study. Additionally, applications of these algorithms to the Pacific OCS wind resource are sparse due to the historical data limitations and previous lack of research motivation.
- Comparison of these ML algorithms against a standard physical law-based approach offers new insights into the most suitable current methods for Pacific OCS wind speed prediction. The ML models are developed by training on one buoy's sea surface-elevation data and testing using the other buoy's data to minimize training location bias and assess the adaptability of each ML algorithm to other offshore locations (Bodini and Optis, 2020). This style of testing helps gauge the greater applicability of these ML methods to other Pacific OCS offshore areas where meteorological data are only available at the sea 100 surface elevation.

### Methods 2

### 2.1 Source Data

Data from the DOE/PNNL lidar buoys is retrieved from the Atmosphere to Electrons Data Archive and Portal, which is supported by the U.S. Department of Energy, Office of Energy Efficiency and Renewable Energy's Wind Energy Technologies 105 Office. The data is publicly available at a2e.energy.gov DOE (2021); Krishnamurthy et al. (2023). All analysis performed in this study uses 10-minute averaged data. For the Humboldt WEA, the data included in this analysis span from October 2020 to January 2022. Due to instrument failure from the end of December 2020 through May 2021, the Humboldt buoy was deployed until July 2022 in order to retrieve a full annual cycle of wind data, but data following January 2022 was not obtained before