# Peer review of "Methods for Pacific Outer Continental Shelf Wind Characterization for Offshore Wind Development"

_Wind Energy Science, 2025_

## Referee Comment (RC2)

**Overall Summary and Impressions for the Author**

The manuscript investigates offshore hub-height wind predictions for the U.S. Pacific Outer Continental Shelf, specifically focusing on the Humboldt and Morro Bay Wind Energy Areas (WEAs). The study compares the performance of machine learning (ML) models, including Random Forest (RF), Gaussian Process Regression (GPR), and Long Short-Term Memory (LSTM) neural networks, with the traditional stability-corrected logarithmic law (S-C log law) for predicting hub-height wind speeds at 100 meters above mean sea level (AMSL). Input features for the ML approaches are derived from sea surface measurements and supervised by lidar measurements of winds aloft collected from DOE-deployed buoys. The models are trained on data from one location and tested on data from the other, a method used in other wind speed extrapolation ML models (Bodini & Optis 2020) to minimize bias and simulate realistic applications. The study reveals that the S-C log law and LSTM models have the best prediction accuracy. ML models advantage over the physical algorithm is the capability to generate additional insights, such as predicting turbulence intensity (TI) and providing confidence intervals in the case of the GPR model. Additionally, challenges like computational expense in algorithms and the influence of atmospheric stability conditions and location on model prediction accuracy are addressed.

The study introduces a comparison of physical modeling and ML-based wind prediction aloft, which appears valuable for advancing offshore wind energy applications. Overall, I think providing additional clarity and explanation of the author's methodology and results would dramatically improve the strength of this manuscript. I would like to see the results section strengthened to showcase the benefits of the ML models. While I think my comments are minor overall, I would recommend major revisions to offer the authors more time to address comments.

**Major Comments**

Line 95 & overall comment- the idea of training and testing the models at different data locations is intriguing, but if data limitation was a big limitation in the study, why not create one more robust model trained and tested on data from both sites to ensure universality using methods such as k-fold cross validation (suitable for small datasets) or time series cross validation (good for including neighbor information)? The authors may want to try out this methodology and compare to their current models. At minimum, can the authors explain why this methodology wasn't explored? Why also did the authors create 6 models

from monthly data instead of one model that includes data from each month? I think future analysis could benefit from incorporating other methodologies.

Line 130 & general comment to address- authors discuss filling nan values with mean hourly values. Can the authors discuss what biases they might have introduced to the data? Another question. LSTM and GPR models use neighbor information, where this nan filling may have been required, but RF doesn't. Did all 3 models use the same input dataset or were the filled values excluded from the RF dataset? Also please explicitly state the size of the datasets. This comment also feeds into a later experiment described in this manuscript. The authors trained at one site and tested at another, and vice versa. However, the Humbolt dataset was much smaller in size. For comparing these models, were they each trained on the same size dataset?

Line 134- What values did you clip the data to? And did it differ between the two sites?

Line 166 and general comment- In my opinion, the section on the atmospheric stability calculations came out of no where. It wasn't mentioned in the abstract or introduction. This part of your analysis needs to be explicitly stated to help guide your reader.

Table 4 & general comment- What values did your tuned algorithm use? You tuned between the range of values given, but what did you use for your final model? General comment, in the manuscript, the author mentions many times that further tuning could improve the models, but this table along with this repeated comment gives the impression the author didn't put significant time into tuning the model. Presumably the parameters underwent rigorous tuning and the best model was used for analysis.

Line 245- Is the same scaling used for the GPR and RF? If not, why?

Figure 7- could you include a fourth subplot showing the input data? At the very least, surface wind speed and air-sea temperature difference, as they were indicated to be the two most important featured variables. Could the authors comment on this case why all three models underpredict wind speed compared to the observations and describe the atmospheric conditions during this period?

Line 325- Can you provide some statistics of performance based on different conditions? For example, accuracy day versus night, stable versus unstable. This would make your results more robust rather than looking at a timeseries representing one day.

Figure 10 and line 342 and 352- I would be interested to see figure 10 replicated for the same location train-test models, perhaps in a supplementary section.

General comment 3.1.3 Overall results show that the LSTM does on par with the S-C log law. However, I'm curious if there are specific cases in which the ML models perform

better. A demonstration of this phenomena would strengthen the paper's conclusion. Perhaps the authors could include a sample time series with the surface data showing the onset of a cold or warm front if it shows that the ML models are more adaptable to forecasting the changes aloft. I recognize this analysis may be beyond the scope of this paper.

Line 377- Have the authors considered a neural network with input channel dropout layers to improve the model's elasticity with missing data?

Line 392- Can you use the lidar's data to confirm sheer conditions?

General comment for analysis based on figure 2- Could you compare the distributions of windspeed from the ML models and physical algorithms?

General comment for analysis based on section 3.1.5- Out of curiosity, could you compute error metrics but separate it by wind speed? Maybe the ML models excel under different wind conditions.

**Minor Comments**

Line 19- Paraphrasing your abstract, you say ML techniques... can be used to predict other wind parameters (plural), but in your paper the only one you evaluate is turbulence intensity. I would be explicit here to not overrepresent your results.

Line 90- provide citation

Line 198- cite the modified log law

Section 2.5- for clarity can you state the lidar buoy data is used as the supervised output dataset for training the model and the S-C log law result is for comparison?

Line 315- describe EMD for general audience

Line 357- Regarding improvement of LSTM over S-C log law, can the author add numbers here?

Line 367- state accuracy

Line 431- I had trouble understanding this sentence. Were the surface variables significant or did feature importance show they were all insignificant?

Line 486- mention this limitation earlier when describing the dataset

---

## Author Comment (AC1)

**Response to reviews**

**Reviewer 1**

This study investigated the accuracy of estimating wind speeds at hub heights from wind speeds on the surface using some methods of machine learning. Below are some comments.

We thank the reviewer for the constructive comments.

**Major comments**

Why does the author only show figures for certain periods? For example, Figure 7 only shows data for the six days from 30 July to 4 August 2021. Interpretations should be made based on all results including other periods as well. Based on this figure alone, it cannot be concluded that the fluctuations in the hourly wind speed predictions of the three MK methods are well represented.

We added text to clarify in the beginning of this section *3.1.2 ML Prediction Samples* that the analysis is based on all results and statistics across all prediction time periods, and that we're just showing an illustrative example of this point in Figure 7.

Additionally, the author concludes that S-C log law and LSTM show the best performance, but in Figure 10, the bias of S-C log law is greater than that of the others, and the reasoning behind this conclusion is unclear.

This is a conclusion based on evaluation of several metrics, not just bias. LSTM produces more accurate wind speed results than the S-C log law for several of the error metrics considered. But RF and GPR show worse performance than the S-C log law for all metrics except bias. There is no single method that had the best performance (in terms of error metrics) across the board, and there is not an objective or industry-standard way to singularly rank these prediction methods "best to worse" based on their individual error metric trade-offs & tendencies.

Furthermore, it is obvious that training at the same location will result in higher accuracy than training at other locations. The purpose of this analysis is unclear.

Using the same train-test location will of course create better results– but the degree to which these results are better, and how much it affects each method's individual error metrics, is not obvious. Our main focus is training and testing between the two separate locations (over a distance of 631 kilometers) in order to simulate realistic industry applications. It is possible,

however, that temporary access to a lidar could allow data collection at a site so that training would be needed for the same location, where the model could be used for future predictions.

Minor comments

* P.1 l.8, AMSL might be above mean sea surface. When the full version of a term first appears in a sentence in the text, place the abbreviation in parentheses after it.

Fixed

* Page 16, Figure 8 should be placed later than wrote in the text.

This figure currently appears on the same page as the text referring to it. We expect the placement will be updated during the typesetting process.

* Page 15, line 306, the author wrote that most similar in performance to the S-C log law are LSTM and GPR, but it should be made clear from which part of the report this can be said.

This statement refers to Figure 8, where it can be seen that RF presents a smoothed time series that deviates more from the observations and the log law. LSTM and GPR both follow the pattern of the S-C log law more closely.

* Page 16, Figure 9 should be also placed later than wrote in the text.

See response for Figure 8 above.

* Page 23, Figure 14 should be also placed later than wrote in the text.

See response for Figure 8 above.

**Reviewer 2**

**Overall Summary and Impressions for the Author**

The manuscript investigates offshore hub-height wind predictions for the U.S. Pacific Outer Continental Shelf, specifically focusing on the Humboldt and Morro Bay Wind Energy Areas (WEAs). The study compares the performance of machine learning (ML) models, including Random Forest (RF), Gaussian Process Regression (GPR), and Long Short-Term Memory (LSTM) neural networks, with the traditional stability-corrected logarithmic law (S-C log law) for predicting hub-height wind speeds at 100 meters above mean sea level (AMSL). Input features for the ML approaches are derived from sea surface measurements and supervised by lidar measurements of winds aloft collected from DOE-deployed buoys. The models are trained on data from one location and tested on data from the other, a method used in other wind speed extrapolation ML models (Bodini & Optis 2020) to minimize bias and simulate realistic applications. The study reveals that the S-C log law and LSTM models have the best prediction accuracy. ML models advantage over the physical algorithm is the capability to generate additional insights, such as predicting turbulence intensity (TI) and providing confidence intervals in the case of the GPR model. Additionally, challenges like computational expense in algorithms and the influence of atmospheric stability conditions and location on model prediction accuracy are addressed.

The study introduces a comparison of physical modeling and ML-based wind prediction aloft, which appears valuable for advancing offshore wind energy applications. Overall, I think providing additional clarity and explanation of the author's methodology and results would dramatically improve the strength of this manuscript. I would like to see the results section strengthened to showcase the benefits of the ML models. While I think my comments are minor overall, I would recommend major revisions to offer the authors more time to address comments.

We thank the reviewer for the constructive comments.

**Major Comments**
Line 95 & overall comment- the idea of training and testing the models at different data locations is intriguing, but if data limitation was a big limitation in the study, why not create one more robust model trained and tested on data from both sites to ensure universality using methods such as k-fold cross validation (suitable for small datasets) or time series cross validation (good for including neighbor information)? The authors may want to try out this methodology and compare to their current models. At minimum, can the authors explain why this methodology wasn't explored?

We thank the reviewer for this idea. Our strategy was "to eliminate model bias to the same train-test location and to simulate a realistic application of the model to a different location." (line 219) The main point of this study is to focus on realistic applications, in which limited spatial and

temporal collected data are major factors in the developing offshore wind industry. Our goal was to compare commonly-accessible ML techniques, for which training "one robust model" is not usually an option. The study used six different train-test datasets and compiled error statistics from each. Using timeseries cross validation within each of these datasets could further refine the error metrics, but at great additional work for potentially not much benefit.

Why also did the authors create 6 models from monthly data instead of one model that includes data from each month? I think future analysis could benefit from incorporating other methodologies.

Monthly data was used because of limitations in data available at both locations. Chronology is important for this application (so assembling selected training data from every month confounds results/error metrics). Also LSTM requires complete data. The text around Line 160 indicates the advantages of short-term data given the differences in seasonal patterns at the two locations. This is also discussed near the end of section 3.1.4, where a comparison is made between 1-month and 3-month training datasets. We agree that future studies should explore additional methodologies. We also added a note in the text referring the reader to section 3.1.4.

Line 130 & general comment to address- authors discuss filling nan values with mean hourly values. Can the authors discuss what biases they might have introduced to the data?

Data filling or imputation is a necessary step when using LSTM which requires complete data. We chose the mean hourly value within the month to fill the missing value. Given that only 1-2% of values were missing for the months selected, we expect the biases introduced to be minimal, but there is no good way to estimate these. Even if the filled data lead to slight increases in prediction error, this would be for all methods tested. Our focus is on a comparison of the three ML and log-law methods, all using the same dataset inputs.

Another question. LSTM and GPR models use neighbor information, where this nan filling may have been required, but RF doesn't. Did all 3 models use the same input dataset or were the filled values excluded from the RF dataset?

Yes, Line 130 states: "Though random forest and Gaussian process regression do not technically require fully-complete datasets, the same filled datasets are used for all ML methods due to the complete-data requirement of LSTM.

Also please explicitly state the size of the datasets. This comment also feeds into a later experiment described in this manuscript. The authors trained at one site and tested at another, and vice versa. However, the Humboldt dataset was much smaller in size. For comparing these models, were they each trained on the same size dataset?

The data availability is different for the two locations, but the actual selected datasets for the two locations are for the same dates. The details and sizes are given in Table 2.

Line 134- What values did you clip the data to? And did it differ between the two sites?

There is no clipping applied. We have added this to the text: "The selected date ranges of the input datasets exclude the few instances of extreme outliers in the source data that were likely attributable to instrument error." The process of scaling to unit variance as described in the text helps to prevent features with much higher variance and/or wider value ranges in their original data from disproportionately influencing the ML model prediction. StandardScaler is used to transform the data of each input feature to z-scores, which have a mean of 0 and a standard deviation of 1. The ranges of the resulting z-score values are centered on 0 and only differ slightly depending on the normality of each dataset.

Line 166 and general comment- In my opinion, the section on the atmospheric stability calculations came out of no where. It wasn't mentioned in the abstract or introduction. This part of your analysis needs to be explicitly stated to help guide your reader.

Thanks for this feedback. We added a note in the intro and in the methods section indicating that stability classification is coming up.

Table 4 & general comment - What values did your tuned algorithm use? You tuned between the range of values given, but what did you use for your final model? General comment, in the manuscript, the author mentions many times that further tuning could improve the models, but this table along with this repeated comment gives the impression the author didn't put significant time into tuning the model. Presumably the parameters underwent rigorous tuning and the best model was used for analysis.

Hyperparameter optimization is performed using automated functions in the cited Python modules in every training step – meaning for each of the six train-test datasets, the hyperparameters of the corresponding RF or GPR model may differ between them. These outputs are not stored because the specific auto-optimized hyperparameters are not important for anyone who wants to replicate the study as these will be selected by the model for any new training dataset.

Comments about further tuning apply mostly for LSTM, for which the hyperparameters are not as easily auto-optimized in the module, and thus were decided on via empirical testing and comparing observations of the loss function and runtime. The final hyperparameters used for LSTM across all datasets are explicitly stated in the text (Section 2.5.3). Again, these parameters for RF and GPR are dataset-dependent and auto-determined by the model.

Our comments about the possibility of further optimization are general, in that there is always a possibility of more optimization rounds with any model, to create slightly better parameter combinations and prediction results, but not necessary for this model comparison study, and perhaps not realistic for a practical application.

Line 245- Is the same scaling used for the GPR and RF? If not, why?

Scaling does not need to be done for RF in the same way that Gaussian scaling is an input data assumption for GPR. RF does not make any assumptions about a normal (Gaussian) distribution of the data. RF is non-parametric and can handle non-linear relationships and data from various distributions without needing to normalize the data. In other words, Gaussian scaling of input data is a specific component of the GPR method, and it would be extraneous to apply this into the typical workflow of the RF method.

Figure 7- could you include a fourth subplot showing the input data? At the very least, surface wind speed and air-sea temperature difference, as they were indicated to be the two most important featured variables. Could the authors comment on this case why all three models underpredict wind speed compared to the observations and describe the atmospheric conditions during this period?

The input data is a month long and so we think it would be difficult to show here in a useful format. The general underprediction shown in *Figure 7* is likely because hub-height wind speed patterns trended slightly higher (in this particular slice shown from Humboldt) than predicted from the typical relationship of hub-height wind speed to surface variables in the training data (from Morro Bay), again emphasizing that this relationship was assessed by each ML method from a location nearly 400 miles away. We have added a comment about this to the text in Sec 3.1.2 (line 299). We also now point the reader to Section 3.1.5 and *Figure 13* for a better and more complete understanding of how atmospheric stability conditions tend to bias results.

Line 325- Can you provide some statistics of performance based on different conditions? For example, accuracy day versus night, stable versus unstable. This would make your results more robust rather than looking at a timeseries representing one day.

Figure 9 is the diurnal average from all 6 months of testing data combined (as stated in the figure's caption), with day and night shown with different shading. Figure 13 also draws its atmospheric stability-based error metrics from all 6 months of testing data. We have added a line at the end of that paragraph to direct the reader to these results.

Figure 10 and line 342 and 352- I would be interested to see figure 10 replicated for the same location train-test models, perhaps in a supplementary section.

Yes, the error metrics for all the train-test models, including same location pairings, are given in Table 5.3 of Chang (2022) as mentioned at the beginning of Section 3.1.4. The text has been updated in a few places to more clearly direct the reader to the additional analysis available in Chang (2022).

General comment 3.1.3 Overall results show that the LSTM does on par with the S-C log law. However, I'm curious if there are specific cases in which the ML models perform better. A demonstration of this phenomena would strengthen the paper's conclusion.

Perhaps the authors could include a sample time series with the surface data showing the onset of a cold or warm front if it shows that the ML models are more adaptable to forecasting the changes aloft. I recognize this analysis may be beyond the scope of this Paper.

Yes, LSTM does better than S-C log law when trained in the same location. In the conclusion we write "When trained and tested in the same location, LSTM surpassed the predictive performance of the S-C log law" and "LSTM is demonstrated to be the most accurate and adaptable ML method for offshore wind speed prediction out of the techniques considered." In real-world practice/applications, on a scale of 0 to ~400 miles, train-test distances are much more likely to be closer to the '0 miles' end of the scale.

Line 377- Have the authors considered a neural network with input channel dropout layers to improve the model's elasticity with missing data?

A dropout algorithm was used in LSTM model training, but the particular method was not employed globally to improve 'elasticity with missing data'. Thank you for the suggestion, which we will consider for future work.

Line 392- Can you use the lidar's data to confirm sheer conditions?

Yes, the frequency distribution of vertical wind shear does differ between the two sites, as shown in Fig 5.5. of Chang (2022). We have added a reference to this figure in the text.

General comment for analysis based on figure 2- Could you compare the distributions of windspeed from the ML models and physical algorithms?

We thank the reviewer for this suggestion. We have added it to the conclusions as a potential further analysis method.

General comment for analysis based on section 3.1.5- Out of curiosity, could you compute error metrics but separate it by wind speed? Maybe the ML models excel under different wind conditions.

We thank the reviewer for this suggestion. We have added it to the conclusions as a potential further analysis method.

**Minor Comments**
Line 19- Paraphrasing your abstract, you say ML techniques… can be used to predict other wind parameters (plural), but in your paper the only one you evaluate is turbulence intensity. I would be explicit here to not overrepresent your results.

This line was modified to read: "These ML techniques are more adaptable for wind energy purposes than conventional physical extrapolation laws, as they *have strong potential to* be used to predict other wind parameters (e.g. turbulence intensity, as presented here) and generate short-term forecasts."

Line 90- provide citation

We have added several citations for each ML method and explicitly referred to Section 3.4 from Chang (2022) for further details. (For brevity we have not included all the details of these references in the manuscript.)

Line 198- cite the modified log law

Done.

Section 2.5- for clarity can you state the lidar buoy data is used as the supervised output dataset for training the model and the S-C log law result is for comparison?

Thank you, we have added clarification to the intro to Section 2.5.

Line 315- describe EMD for general audience

Done.

Line 357- Regarding improvement of LSTM over S-C log law, can the author add numbers here?

We have added a reference to Section 3.1.4 in the text where all the numbers and comparison are discussed in more detail.

Line 367- state accuracy

This line already includes the accuracy: "These conditions produce RMSE of 1.19 m/s and 0.89 m/s for the Humboldt and Morro Bay WEA locations respectively, which are the lowest RMSE values obtained from any prediction method used in this study, including the S-C log law."

Line 431- I had trouble understanding this sentence. Were the surface variables significant or did feature importance show they were all insignificant?

Analysis and selection of input features (as mentioned was done for the hub-height wind speeds in Line 115) was not investigated for TI – in other words, the same selection of inputs for hub-height wind speed (Table 1) were copied and used as predictor variables for TI. We have edited the text to more clearly say that TI might have other input variables that would help improve predictions for it specifically.

Line 486- mention this limitation earlier when describing the dataset

We edited the text to mention the exclusion of wind direction and humidity data in Section 2.1.